# Developmental Stages-Specific Response of Anise Plants to Laser-Induced Growth, Nutrients Accumulation, and Essential Oil Metabolism

**DOI:** 10.3390/plants10122591

**Published:** 2021-11-26

**Authors:** Mohammad K. Okla, Mohamed Abdel-Mawgoud, Saud A. Alamri, Zahid Khorshid Abbas, Wahidah H. Al-Qahtani, Salem Mesfir Al-Qahtani, Nadi Awad Al-Harbi, Abdelrahim H. A. Hassan, Samy Selim, Mohammed H. Alruhaili, Hamada AbdElgawad

**Affiliations:** 1Botany and Microbiology Department, College of Science, King Saud University, P.O. Box 2455, Riyadh 11451, Saudi Arabia; Okla99@yahoo.com (M.K.O.); Salamri@hotmail.com (S.A.A.); 2Department of Medicinal and Aromatic Plants, Desert Research Centre, Cairo 11753, Egypt; 3Biology Department, College of Science, Tabuk University, Tabuk 71491, Saudi Arabia; Znourabbas@ut.edu.sa; 4Department of Food Sciences & Nutrition, College of Food & Agriculture Sciences, King Saud University, Riyadh 11451, Saudi Arabia; wahida@ksu.edu.sa; 5Biology Department, University College of Taymma, Tabuk University, P.O. Box 741, Tabuk 47512, Saudi Arabia; salghtani@ut.edu.sa (S.M.A.-Q.); nalharbi@ut.edu.sa (N.A.A.-H.); 6Department of Food Safety and Technology, Faculty of Veterinary Medicine, Beni-Suef University, Beni Suef 62511, Egypt; abdelrahim@vet.bsu.edu.eg; 7Department of Clinical Laboratory Sciences, College of Applied Medical Sciences, Jouf University, Sakaka 72341, Saudi Arabia; sabdulsalam@ju.edu.sa; 8Department of Medical Microbiology and Parasitology, Faculty of Medicine, King Abdulaziz University, Jeddah 21589, Saudi Arabia; malruhaili@kau.edu.sa; 9Botany and Microbiology Department, Faculty of Science, Beni-Suef University, Beni Suef 62521, Egypt; hamada.abdelgawad@uantwerpen.be; 10Integrated Molecular Plant Physiology Research, Department of Biology, University of Antwerp, 2020 Antwerpen, Belgium

**Keywords:** anise sprouts, mature plants, He–Ne laser, nutritious metabolites, antioxidant

## Abstract

Compared to seeds and mature tissues, sprouts are well known for their higher nutritive and biological values. Fruits of *Pimpinella anisum* (anise) are extensively consumed as food additives; however, the sprouting-induced changes in their nutritious metabolites are hardly studied. Herein, we investigated the bioactive metabolites, phytochemicals, and antioxidant properties of fruits, sprouts (9-day-old), and mature tissue (5-week-old) of anise under laser irradiation treatment (He-Ne laser, 632 nm). Laser treatment increased biomass accumulation of both anise sprouts and mature plants. Bioactive primary (e.g., proteins and sugars) and secondary metabolites (e.g., phenolic compounds), as well as mineral levels, were significantly enhanced by sprouting and/or laser light treatment. Meanwhile, laser light has improved the levels of essential oils and their related precursors (e.g., phenylalanine), as well as enzyme activities [e.g., O–methyltransferase and 3-Deoxy-D-arabino-heptulosonate-7-phosphate synthase (DAHPS)] in mature tissues. Moreover, laser light induced higher levels of antioxidant and anti-lipidemic activities in sprouts as compared to fruits and mature tissues. Particularly at the sprouting stage, anise was more responsive to laser light treatment than mature plants.

## 1. Introduction

Sprouts are considered as rich sources of bioactive metabolites such as vitamins, minerals, and polyphenols, with various biological properties such as antioxidant and antitumor activities [1,2]. When compared to seeds and mature plants, sprouts are known to have low levels of anti-nutritional factors, so they are strongly recommended as popular healthy foods [2].

Recently, the application of different techniques for enhancing the nutritional quality of sprouts and mature plants has been attracting much interest [3,4,5]. In this regard, the use of laser irradiation as a biophysical elicitor, has proved to increase plant growth and development, induce high content of phytochemicals and enhance plant productivity [4,6]. Laser light has been divided into two types; pulsed laser, which has been used in medical applications; and continuous laser, which was applied in improving crop production [6,7]. Plants could respond physiologically to laser treatment (light, electromagnetism, and temperature) and their responses are based on the ability of plant macromolecules to absorb a specific wavelength of laser light to trigger photosynthetic activity, which leads to increased growth and biomass accumulation [6]. Previous reports have demonstrated the positive impact of laser light on increasing the quality and accumulation of bioactive metabolites of many plants, such as fennel, coriander, white lupine, faba bean, soybean, lemongrass sprouts, and buckwheat sprouts [4,8,9,10,11].

*Pimpinella* species are belonging to the family *Apiaceae*, which are mostly used as condiments or vegetables, in addition to their use in traditional therapy in many parts of the world [12]. Among the most common *Pimpinella* species, *Pimpinella anisum* (anise) has been extensively consumed as food additives and beverages and is also used in the pharmaceutical industry due to its distinct flavor and therapeutic values [13]. Anise fruits (aniseeds) have been used in dry/fresh form in traditional medicine to cure many ailments [14]. Moreover, aniseeds contain essential oils with a wide range of biological activities, such as antimicrobial, antioxidant, carminativum, anti-spasmoliticum, and antidiabetic activities [15,16,17]. Principally, aniseeds contain *trans*-anethole as a major volatile component [18], in addition to other constituents such as eugenol, anisaldehyde, methylchavicol, coumarins, scopoletin, estragole, umbelliferone, terpene hydrocarbons, estrols, polyenes, and polyacetylenes [17]. The medicinal properties of aniseed are frequently attributed to its content of bioactive molecules such as phenolic components and terpens. Therefore, increasing the bioactive contents of anise sprouts and mature plants by using eco-friendly approaches, such as laser light, could enhance the plant’s nutritive and health-promoting values.

To the best of our knowledge, few studies have dealt with the effect of laser light on aniseeds morphological properties and yield [19]. Thus, the present study aimed at investigating the laser-induced effects on growth, minerals, vitamins, essential oils, as well as total bioactive metabolites and biological activities (antioxidant, anti-lipidemic) of anise sprouts and their mature plants under He-Ne laser treatment in comparison with control. At the same time, we also compared the significant differences between anise sprouts and mature plants, regarding their response to the effect of laser light on their bioactive contents and biological activities.

## 2. Material and Methods

### 2.1. Plant Material and Experimental Conditions

Fruits of anise were collected and obtained from Agricultural Research Centre, Giza, Egypt. The fruits of anise were soaked in distilled water for 2 h, then they were divided into 2 groups; untreated and laser-treated groups (each one contains 100 seeds). Helium-neon (He-Ne) laser system (equipment whitening, laser II, DMC Equipment Ltd., Vista, CA, USA) was used as a light source. The treated group was irradiated by He–Ne laser (632 nm at 5 mW for 5 min and 500 mJ energy from the embryonic area side, beam diameter 1 mm), whereas the distance between laser light and the fruits was 12 cm. The untreated treated fruits were placed in dark covered with a box, under controlled conditions. All these conditions were chosen after a preliminary experiment and our previous studies [4,11]. The experiment was repeated 3 times. The irradiated fruits were rinsed in distilled water and spread on trays and irrigated for the first three days with Milli-Q water, then placed in the dark. Afterward, the fruits were maintained at 25 °C in a growth chamber under a 16 h light/8 h dark cycle through white fluorescent tubes with 400 μmol m^−2^ s^−1^ photosynthetically active radiation (PAR). Thereafter, 100 mL aquaponic water was poured over the fruits in each tray. Mature plants (five weeks-old) were cultivated at 21/18 °C in a custom-built climate-controlled chamber with a 16/8 h day/night photoperiod (150 μmol PAR m^−2^ s^−1^, 60% humidity). After 9 days, the sprouts were taken as fresh weight and then stored at −80 °C for further analyses, where the chemicals and reagents were purchased from Sigma Aldrich. Five biological replicates (each biological replicate was a pooled of ten plants) were used for each measurement

### 2.2. Determination of Photosynthetic Rate

Photosynthesis (μmol CO_2_ m^−2^ s^−1^) was determined according to [5], by using EGM-4 infrared gas analyzer (PP Systems, Hitchin, UK). The photosynthetic rate was evaluated from 180 s measurements of net CO_2_ exchange (NE).

### 2.3. Pigment Analysis

Anise sprouts and mature plants were homogenized at 7000 rpm for 1 min in acetone by using a MagNALyser (Vilvoorde, Belgium), then centrifuged for 20 min (4 °C, 14,000× *g*). The supernatant was then filtered (Acrodisc GHP filter, 0.45 μm 13 mm). Afterward, HPLC (Shimadzu SIL10-ADvp, reversed-phase, at 4 °C) was used to analyze the obtained solution [4]. Carotenoids were separated on a silica-based C18 column (Waters Spherisorb, 5 μm ODS1, 4.6 × 250 mm, Agilent Technologies, Santa Clara, CA, USA), whereas two solvents were used; (A) acetonitrile: methanol: water (81:9:10) and (B) methanol: ethyl acetate (68:32). Chlorophyll a,b, and β-carotene were extracted, and then quantified using a diode-array detector (Shimadzu SPDM10Avp, Japan, Tokyo) at four wavelengths (420, 440, 462, and 660 nm).

### 2.4. Elemental Analysis

Elemental analysis was done, using inductively coupled plasma (ICP-MS, Finnigan Element XR, Scientific, Bremen, Germany). Macro and micro-elements were evaluated according to [5]. About 200 mg from both treated and control non-treated sprouts and mature plants were digested in HNO_3_/H_2_O solution (5:1 *v/v*) in an oven. Then, the concentrations of macro-minerals and trace elements were estimated at 25 °C by using inductively coupled plasma mass spectrometry (ICP-MS, Finnigan Element XR, and Scientific, Bremen, Germany), where nitric acid in 1% was used as a standard.

### 2.5. Vitamins Analysis

Tocopherols and ascorbate and were determined according to [4] using reversed-phase HPLC, where tocopherols and ascorbate were mixed with hexane and metaphosphoric acid in MagNALyser, respectively. Then, centrifugation was done (4 °C, 14,000× *g*, 20 min). Meanwhile, thiamine and riboflavin were separated on a reverse-phase (C18, Agilent Technologies, Santa Clara, CA, USA) column (HPLC, methanol: water) [4].

### 2.6. Essential oil Analysis

#### 2.6.1. GC-MS Analysis

The tested sprouts and mature plants were air-dried, and then extraction of essential oil was done using 15 g. The dried parts were subjected to steam distillation for three hours using a Clevenger-type instrument. The essential oil contents were determined by using GC/MS according to [20]. GC–MS analysis was done by using a Thermoquest GC–MS instrument (EI mode at 70 eV), where a DB-1 fused silica capillary column (60 m * 0.25 mm i.d., film thickness 0.25 mm) was used. The temperature was increased from 40 to 250 °C at 4 °C min^−1^, then kept constant at 250 °C for 10 min, while the temperatures of detector and injector were 300 °C and 250 °C, respectively. The carrier gas used was Helium (at a flow rate of 1.1 mL min^−1^. Identification of the detected compounds was achieved by comparing their mass spectra with those obtained from NIST library. For the retention index (RI) of identified essential oil, please check Appendix A.

#### 2.6.2. Determination of DAHPS

Determination of DAHPS activity was done according to [21]. Fresh samples were homogenized in 3 mL pre-cooled Tris-HCL buffer (50 mM, pH = 7.4) containing polyvinylpyrrolidone (1%), phenylmethylsulfonyl fluoride (0.1 mM), leupeptin (10 μM), and 2-mercaptoethanol (1.4 mM), The mixture was kept at 4 °C for 30 min, then centrifugation was done (12000, 20 min). The supernatant was used for the assay, whereas the mixture contained the extract (0.8 mL), Tris-HCl buffer (2.2 mL, 50 mM, pH = 7.5), phosphoenolpyruvate (0.2 mM), 0.1 mM MnSO_4_/0.1 mM CoCl_2_, and erythrose-4- phosphate (0.1 mM). Then, incubation was done at 30 °C for 30 min. Afterward, the reaction was started by adding the enzyme, then terminated by adding 500 μL trichloroacetic acid (25%), while the control was prepared without the enzyme. The enzyme activity was detected on the basis of the amount of enzyme used for the synthesis of 1 nmol of DAHPS per minute at 30°C. Finally, the concentration of DAHPS was detected at 549 nm.

#### 2.6.3. Determination of PAL

For determination of PAL activity, fresh samples were homogenized in 3 mL pre-cooled sodium borate buffer (0.1 M, pH = 8.8) containing polyvinylpyrrolidone (0.4%), EDTA (1 mM), 2-mercaptoethanol (5 mM), The mixture was kept at 4 °C for 30 min, then centrifugation was done (12000, 20 min). The supernatant was used for the assay, whereas the mixture contained the extract (0.8 mL), 2.2 mL sodium borate buffer (0.1 M, pH = 8.8) containing L-Phe (120 μM). Then, incubation was done at 25 °C for 40 min. Afterward, the reaction was terminated by adding 120 μL HCL (6 N), while the control was prepared without the enzyme. Detection of the product, trans-cinnamic acid, was done at 290 nm. The enzyme activity was detected on the basis of deamination of 1.0 nmol of L-phenylalanine to cinnamic acid per minute.

### 2.7. Determination of Phenolic Profile

#### 2.7.1. Determination of Total Phenolic Content

Extraction of flavonoids and phenolic acids was done by homogenizing 100 mg of sprouts in 1 mL of 80% ethanol. Centrifugation was done for 20 min at 4 °C, and then the flavonoid content was measured by using the modified aluminum chloride colorimetric method (quercetin as a standard). Meanwhile, the total phenolic content was estimated by using a Folin–Ciocalteu assay (gallic acid as a standard) [4].

#### 2.7.2. HPLC Analysis

The Individual flavonoids and phenolic acids were detected by homogenizing 50 mg of samples in acetone–water (4:1) for 24 h. Thereafter, phenolic compounds have been analyzed by using HPLC (SCL-10A vp, Shimadzu Corporation, Kyoto, Japan), equipped with a Lichrosorb Si-60, 7 μm, 3 × 150 mm column, diode array detector), whereas 3,5-dichloro-4-hydroxybenzoic was used as an internal standard. The mobile phase consisted of water-formic acid (90:10), and acetonitrile/water/formic acid (85:10:5). The detection of each compound’s concentration was done using a calibration curve of the corresponding standard. For the retention index (RI) of identified essential oil, please check Appendix A.

### 2.8. Biological Activities

#### 2.8.1. Antioxidant Activities

According to the methods outlined in [4], the antioxidant potential of the examined sprouts and plants was evaluated through different assays; i.e., diphenylpicrylhydrazyl (DPPH), ferric reducing antioxidant power (FRAP), oxygen radical absorbance capacity (ORAC), and inhibition of LDL (low-density lipoprotein) oxidation (TBARS and conjugated dienes) [22,23,24]. For determination of FRAP activity, FRAP reagent (180 μL) and ethanol extracts (20 μL) were added into each well in the micro-plate, then incubation was done for 30 min at 37 °C. The absorbance was detected at 593 nm with a microplate reader (Synergy Mx, Biotek Instruments Inc., Vermont, VT, USA). Trolox was used as a standard. For determination of ORAC, 120 μL of fluorescein (112 nM) dissolved in phosphate buffer (75 mM) was added into the micro-plate. Afterward, 20 μL of the samples, 20 μL of phosphate buffer (blank) and 20 μL of trolox (standard) were added to the plate, then incubation was done for 15 min at 37 °C, and the absorbance was detected at 485/520 nm. Thereafter, 80 μL of AAPH (62 mM) (2,2′-azobis 2-methylpropionamidin dihydrochloride) was added, and then, the absorbance was detected at 485/520 nm. The difference between the two measurements was detected using a standard row. For determination of LDL, dialyzed LDL (100 µg protein/mL) was diluted in 10 mM PBS (phosphate-buffered saline containing 0.01 M phosphate-buffer and 0.15 M NaCl, pH 7.4), and incubation was done at 37 °C with or without 10 µM CuSO_4_. Afterward, oxidation was done in the presence or absence of colostrum proteins, and then lipid peroxidation was detected. Thiobarbituric acid reactive substances (TBARS) were determined at 532 nm/600 nm, where 1,1,3,3-Tetramethoxypropane was used as a standard. Conjugated diene was detected at 232 nm of LDL solution (100 µg protein/mL) in PBS incubated with CuSO_4_ (10 µM) in the presence or absence of different concentrations of bovine colostrums protein.

#### 2.8.2. Hypocholesterolaemic Activity

##### Inhibition of Micellar Solubility of Cholesterol

In order to evaluate the effect of the tested sprouts and plants on micellar solubility of cholesterol, the protocol described in [3] was used, where the extracts were added to micellar solution [15 mM sodium phosphate, 10 mM sodium taurocholate, 2 mM cholesterol, 5 mM oleic acid, 132 mM NaCl]. Afterward, the mixture was sonicated for 2 min and incubated in a water bath at 37 °C for 24 h. Thereafter, centrifugation was done for 60 min (40,000 rpm, 20 °C). The cholesterol content was evaluated at 500 nm by using a cholesterol analysis kit (Pointe Scientific, New York, NY, USA, C7510).

##### Pancreatic Lipase Inhibition Assay

The method described in [3] was used to evaluate the impact of the tested extracts on pancreatic lipase by using 4- MUO as a substrate. About 0.5 mL of the extracts were taken and mixed with 0.5 mL of lipase. Centrifugation was done for 10 min at 4000 rpm, and then 2 mL of the 4-MUO solutions were added. Incubation was done at 37 °C. Aliquots of 0.2 mL were taken at different time points, and 4-MUO hydrolysis by lipase was detected at 350/450 nm. A logarithmic regression curve was created to detect IC50 values (mg/mL).

### 2.9. Statistical Analyses

To perform the statistical analyses, the SPSS statistical package (SPSS Inc., Chicago, IL, USA) was used. Each experiment was replicated at least two times, and for all assays, 3 to 5 replicates were used, and each replicate corresponded to a group of sprouts and mature plants harvested from a certain tray. A one-Way Analysis of Variance (ANOVA) test was applied. Tukey’s test was used as the post-hoc test for separation of means (*p* < 0.05). Cluster analysis was performed by using Pearson distance metric of the MultiExperiment Viewer (MeV)™ 4 software package (version 4.5, Dana-Farber Cancer Institute, Boston, MA, USA).

## 3. Results and Discussion

### 3.1. Physical Properties and Biomass Accumulation of Anise Fruits, Sprouts, and Mature Plants as Affected by Laser Light Treatment

Laser light has been known to act as a stimulating factor for plant growth and yield, as well as the sprouting process [4,6]. The absorption of laser light at a certain wavelength could activate the photosynthesis process and increase seed internal energy, which consequently accelerates cell division and improve enzymatic activities [4,6,10]. This might also include the induction of some phytohormones, such as indole-3-acetic acid (IAA), which is important for cell division and growth in germinating seeds [8]. Such laser-provoked changes in plants could eventually lead to improved growth and biomass accumulation.

Supporting such a hypothesis, the current results have shown significant increases in length, width, thickness, pod length, and yield of laser-treated anise fruits, when compared to their respective controls (Figure 1). Meanwhile, fruit mass was not affected by laser treatment. In agreement, previous studies have demonstrated the enhancing effects of laser on growth and biomass production in many plant species, such as anise, cumin, fennel, coriander, white lupine, faba bean, lemongrass sprouts, and buckwheat sprouts [4,8,9,10,11,19]. Thus, laser light might affect plant growth, either directly through activating germination and thermodynamic parameters, or indirectly by its positive impact on the photosynthetic activity.

The present investigation has also revealed that laser light significantly induced the accumulation of photosynthetic pigments in aniseed sprouts and mature plants, whereby a significant increment was observed in pigment contents of laser-treated sprouts, particularly in the levels of chlorophyll a, b, and (a + b) which were increased by 106, 24 and 77% (Figure 2). Meanwhile, insignificant increments were detected in pigment contents of laser-treated mature plants, compared to their untreated controls. By comparing their response to laser treatment, anise sprouts seemed to better respond, thus accumulating higher photosynthetic pigments than mature plants do. In this regard, the higher chlorophyll content, under the effect of laser light, is expected to trigger the photosynthetic activity, which leads to higher sugar content, and consequently higher biomass and yield accumulation [25]. In addition, it has been reported that the photon energy of laser light could be absorbed by chlorophyll, and hence directly affects the photosynthetic activity [26]. Moreover, several studies have addressed the key role of carotenoids in the photosynthesis process, so the biosynthesis of carotenoids is expected to be influenced by light intensity and quality [27]. Consequently, the production of carotenoids might be triggered by the genes incorporated in carotenoid biosynthesis [28].

In accordance, several reports have dealt with the positive impact of laser light on levels of chlorophyll a and b in some plant species, such as *Isatis indogotica*, sunflower, and soybean, as well as in buckwheat sprouts [4,6,10,29]. Similarly, it has been previously shown that various light sources might exert a positive impact on chlorophyll accumulation in buckwheat sprouts [30].

### 3.2. Sprouting and Laser Light Induced a More Pronounced Effect on Nutritive Values of Anise

The determination of nutritive values of plants has been dependent on their bioactive contents of primary metabolites (e.g., lipid, proteins, and sugars), and secondary metabolites (e.g., phenolic compounds) [5]. The increasing of such bioactive metabolites by using promising approaches, such as laser light, could significantly enhance the plant’s nutritional quality [4].

The present results have revealed that laser light treatment significantly enhanced the amount of some detected bioactive primary (lipid, proteins, and sugars) and secondary metabolites (saponins, steroids) in both sprouts and mature plants, in comparison to untreated controls (Table 1). Meanwhile, the total phenols, flavonoids, and tannins were increased only in anise sprouts. The total nutrients of fruits were almost comparable to those of control plants. So, the laser-treated anise sprouts might respond better to laser effects, hence accumulate higher content of bioactive metabolites, than the mature plants do. The increased sugar content might be associated with the enhanced photosynthetic activity, reported herein. As a result, the bioavailability of sugars could provide the carbon skeleton and energy needed for the biosynthesis of various classes of other metabolites [25,31]. This could explain the increased contents in some bioactive metabolites, reported in our study, such as phenolic compounds, saponins, and steroids. In agreement, laser light has been previously reported to increase total sugar contents in lemongrass sprouts and consequently increased their primary and secondary metabolites [11].

The laser-enhanced bioactive contents could be also attributed to the ability of laser light to activate plant metabolism, which could improve the nutrient status and plant productivity [6,32]. Moreover, sprouting has been recognized as an effective way to enhance the active primary and secondary metabolites, since seeds might be subjected to many physiological changes. For example, some bioactive metabolites, e.g., phenolic compounds, have been reported to increase gradually during sprouting in buckwheat species [33].

Sprouts have been used as rich sources of vitamins and mineral elements, whose deficiency could risk human health. Therefore, enhancing the contents of vitamins and minerals in aniseeds sprouts by using promising techniques, such as laser light, might boost their nutritional and health-promoting values. According to our results, the majority of the detected minerals have significantly increased in response to laser treatment in aniseeds sprouts (dominated by P, K, Ca, and Mg), and mature plants (mostly N), whereby anise sprouts and mature plants are likely to equally respond to laser light effects on their mineral contents (Table 1). Meanwhile, the fruits appeared to act as the control plants.

In agreement with our results, previous reports have shown laser light to enhance the levels of mineral elements in many plant species, e.g., N, P, and K in fennel and coriander [9], P and K in anise and cumin [19], K, Ca and Mg in sunflower [29], k in sugar beet [32] and K, P and Na in buckwheat sprouts [4]. The laser-enhanced effects on mineral contents could be explained by the ability of laser light to act as an inducer for more energy production from plant cell which could stimulate plant metabolism and nutrient uptake [34], or by increased root growth which consequently increases mineral uptake [35].

Regarding vitamins, the present investigation also showed that laser light treatment caused significant increases in some of the measured vitamins (mostly Vit A, E, and C in aniseeds sprouts, and Vit A and K in the mature plant) (Table 1). Laser light appears to have a more pronounced effect on the vitamin content of anise mature plant than that of the sprouts, while the fruits were significantly similar to control plants regarding their vitamin content.

Our results could be supported by recent studies by [4], who demonstrated a significant enhancement in the contents of Vit C, E, B1, and B2 in buckwheat sprouts after laser treatment. Moreover, [10] reported that laser light enhanced Vit C content in soybean. The improved vitamin content in response to laser treatment could be ascribed to activating the photosynthetic process, which consequently leads to increased carbohydrates production that might be used as a precursor for the synthesis of various classes such as vitamins [25]. In addition, the accumulation of vitamins could be also due to increasing the energy provided to the seeds after laser light exposure, which could be then converted into chemical energy and accelerated the metabolic events during the sprouting process [26].

The current investigation has also demonstrated that the chemical profile of anise essential oil contained 18 identified compounds, whereas *trans*-anethole was reported as the dominant component (Table 1). When compared to untreated control, the laser-treated anise sprouts and mature plants have exhibited significantly higher contents of most detected essential oils. Such increases were almost similar in both laser-treated sprouts and mature plants, indicating that they responded equally to laser light effects on their essential oil contents. The fruits were observed to contain almost similar amounts of essential oils to the control plants.

The enhanced levels of such essential oils could be attributed to the increased photosynthetic activity, in response to laser light, which consequently stimulated the sugar content that serves as a precursor for the synthesis of different metabolites, e.g., essential oils [11]. In agreement with our results, laser light has been demonstrated to increase the essential oil contents of lemongrass sprouts [11]. Moreover, it has been previously reported that *trans*-anethole is the main component which comprises 90% of aniseeds essential oil, being the most important constituent responsible for the distinct aroma and taste of aniseed [13,18], in addition to its variable biological properties, e.g., antibacterial, antifungal, antioxidant, and anti-migraine headache effects [36,37,38]. Also, other components, such as anisaldehyde, methyl chavicol, have been previously reported in high amounts in aniseed essential oil [17].

Phenylpropanoid compounds are known to be the main components of essential oils, being synthesized from phenylalanine through the cinnamic and shikimic acids pathway [21,39]. In this regard, several enzymes are involved in these biosynthetic pathways, such as 3-deoxy-D-arabinoheptulosonate-7-phosphate synthase (DAHPS) which is incorporated in the shikimic acids pathway, whereas shikimic acid is being converted into chorismite (a precursor for phenylalanine) [40]. Then, phenylalanine is converted into cinnamic acid with the aid of the phenylalanine aminolyase enzyme (PAL). Eventually, p-coumaric acid is synthesized from cinnamic acid. Both p-coumaric and cinnamic acids are required for the biosynthesis of essential oils in plants [21,41].

Thus, in order to get insight into the laser-induced changes in essential oils precursors and their related biosynthetic enzymes, the effect of laser on the levels of phenylalanine, PAL, DAHPS, cinnamic acid, and shikimic acid in anise sprouts and mature plants was investigated. The obtained results have shown that laser light treatment led to significant increments in phenylalanine, PAL, DAHPS, cinnamic acid, and shikimic acid in both sprouts and mature plants, when compared with their respective controls (Table 1). The increases in DAHPS, cinnamic acid, and shikimic acid were almost similar in both laser-treated sprouts and mature plants, while the increases in phenylalanine and PAL were more obvious in laser-treated sprouts than the mature plants, indicating that the effect of laser on essential oils precursors was more pronounced in sprouts than the mature plants. The fruits were observed to contain almost similar amounts of essential oils to the control plants. Similar to our results, eCO2 has been previously reported to exert positive effects on different stages of anise plants [42]. The obtained results could indicate the enhancement in essential oil metabolism in different stages of anise plants, exposed to laser irradiation treatment.

So, improving the essential oil content of anise sprouts and mature plants by using laser light could increase their content of anethole, thus enhancing their quality and therapeutic values, particularly antioxidant properties.

### 3.3. Laser Light Increased the Antioxidant Potential of Anise Sprouts and Mature Plants by Enhancing Their Phenolic Content

The antioxidant activities of plants have been mostly associated with their higher content of phenolic compounds (flavonoids and phenolic acids) [43]. Therefore, the enhancement of phenolic content of plants by application of biophysical methods, such as laser light, could effectively increase their antioxidant capacities [4]. In the present investigation, gallic and caffeic acids had the highest amount among detected phenolic acids, while quercetin and naringenin were the major flavonoids reported in both anise sprouts and mature plants (Table 2). When exposed to laser light treatment, phenolic and flavonoid profiles of anise sprouts and mature plants were positively affected, the effect that was more pronounced on sprouts than mature plants. There were also insignificant differences between the fruits and control plants regarding their phenolic contents.

In line with our findings, previous and recent studies confirmed that aniseeds contain high amounts of phenolic and flavonoid compounds, whereas gallic and caffeic acids were the main phenolic acids [44,45], while naringenin was the predominant flavonoid detected in anise fruits [14]. It was also reported that aniseeds contain high amounts of total phenolic and flavonoids that are the main contributors to antioxidant and other bioactivities of aniseeds [44,45]. Moreover, earlier studies recorded the enhancing effects of laser light on the total phenolic content in some plants such as sunflower, soybean, lemongrass sprouts, and buckwheat sprouts [4,10,11,29]. In a similar context, different light sources, such as light-emitting diodes (LEDs) and UV, have been previously investigated for their positive effects on increasing secondary metabolites biosynthesis, especially flavonoids in buckwheat sprouts [28,46]. The enhanced levels of phenolic and flavonoids might be ascribed to stimulated photosynthetic activity in response to laser light, which consequently increased the sugar content that acts as a precursor for the synthesis of different classes of metabolites, such as phenolic compounds [25,31]. Besides, the increases in flavonoid contents could be also attributed to the enhanced enzymatic activities of phenylalanine ammonia-lyase (PAL) and tyrosine ammonia-lyase (TAL) which are being involved in the phenylpropanoid pathway [47].

As a consequence of increasing their phenolic and essential oil contents under laser light treatment, the tested sprouts and mature plants have exhibited potent antioxidant activities as tested by different assays (DPPH, FRAP, TAC, ORAC, lipid peroxidation, TBARS and conjugated diene, and inhibition % of hemolysis) (Table 3). Both sprouts and mature plants significantly responded to laser light effects on their antioxidant capacities, while the fruits exhibited similar antioxidant capacities to those of control plants.

In this regard, aniseed has been recognized as a rich source of antioxidant constituents [14,17,36]. It was also reported that aniseeds contain high amounts of total phenolic and flavonoids compounds which are the main contributors to antioxidant and other bioactivities of aniseeds [44,45]. For instance, quercetin and naringenin, detected herein, were previously investigated for their strong antioxidant activities [14,48]. In accordance with our results, laser light has been recently found to enhance the DPPH, FRAP, and ABTS antioxidant capacities of both lemongrass and buckwheat sprouts, by increasing their phenolic and flavonoid contents [4,11]. Similarly, the application of other light sources, such as UV irradiation, enhanced the antioxidant activity of buckwheat sprouts [46].

### 3.4. Laser Light Improved the Anti-Lipidemic Activity Particularly in Anise Sprouts

In the present study, the anti-lipidemic activity of aniseed sprouts and mature plants was evaluated by measuring their inhibitory effect on α-amylase, lipase activity, and cholesterol levels, in response to laser light. Results demonstrated superior anti-lipidemic activity after laser treatment in both treated groups (sprouts and mature plants) as indicated by decreasing α-amylase/lipase activities and cholesterol levels, compared to non-irradiated plants (Table 3). Both sprouts and mature plants significantly responded to laser light effects on their anti-lipidemic activities, while the fruits exhibited similar effects to those of control plants.

Hypolipidemic and antidiabetic activities of aniseed were previously reported and proved to be strongly correlated to its antioxidant constituents, such as phenolic compounds [15]. Similar to our results, laser light has been previously found to enhance the ability of lemongrass sprouts to decrease the levels of cholesterol, triglycerides, and low-density lipoprotein (LDL) [11]. Such hypocholesterolemic activity was reported to be associated with the availability of bioactive metabolites such as phenolic compounds, which could help in the excretion of cholesterol in the feces [49]. Moreover, the anti-lipidemic activity of many plants, such as *grewia optiva*, has been ascribed to their phenolic content, particularly quercetin, being responsible for its α-glucosidase and α-amylase activities. Quercetin has also previously exhibited better anti-lipase activity than other phytochemicals [50].

### 3.5. Tissue-Specific Response to Laser Light Treatment

The hierarchical clustering analysis has shown that there was an obvious tissue-specific response to laser light effects (Figure 3). Anise sprouts have been shown to be more responsive to the enhancing effect of laser light, where it had the highest contents of pigments, phenolic compounds, total proteins, sugars, vitamins, and minerals, followed by mature plants. Meanwhile, anise fruits exhibited the lowest response to laser light treatment. The data presented in Figure 3 has also shown that there were 6 clusters, whereas cluster 1 contained the highest amount of some volatile compounds (e.g., sabinene and para-anisaldehyde) which were mainly concentrated in the mature tissues under laser effects. Meanwhile, cluster 2 had the highest content of phenols, Zn, and beta carotene (mainly in the laser-irradiated sprouts). Also, cluster 3 had the highest concentration of tocopherols, ascorbic acid, sugars, crude fibers, saponins, and some minerals (e.g., K, Mg, and Fe), mostly accumulated in the laser treated-sprouts. Cluster 4 and 5 were rich in some phenolic compounds (e.g., kaempferol, chlorogenic acid, and apigenin), and Ca (accumulated in both laser treated-sprouts and mature tissues). Eventually, cluster 6 contained Cu, P, and ferulic acid (concentrated in the laser treated-sprouts). The variations among the three developmental stages might be ascribed to their diversity and ontogeny. Our results could be supported by previous studies that discussed the positive effects of laser on increasing the primary and secondary metabolites of many plants and sprouts such as buckwheat and lemongrass sprouts [4,11]. Other studies have also demonstrated that laser light could induce seed germination and thermodynamic parameters, which would be reflected in increasing photosynthesis and biochemical processes [6,7].

## 4. Conclusions

Based on the above results, it could be concluded that laser light could be an effective technique to enhance the growth and photosynthetic activity of anise sprouts and mature plants. Consequently, laser light increased the total bioactive metabolites as well as the levels of minerals and vitamins, with concomitant increments in antioxidant activities in both anise sprouts and mature plants. Interestingly, laser light effects might be more pronounced on anise sprouts than their mature plants. Thus, this study could support the use of laser light as an advantageous approach to increase the nutritional and health-promoting values of both anise sprouts and mature plants, preferably in the sprouting stage.

## Figures and Tables

**Figure 1 plants-10-02591-f001:**
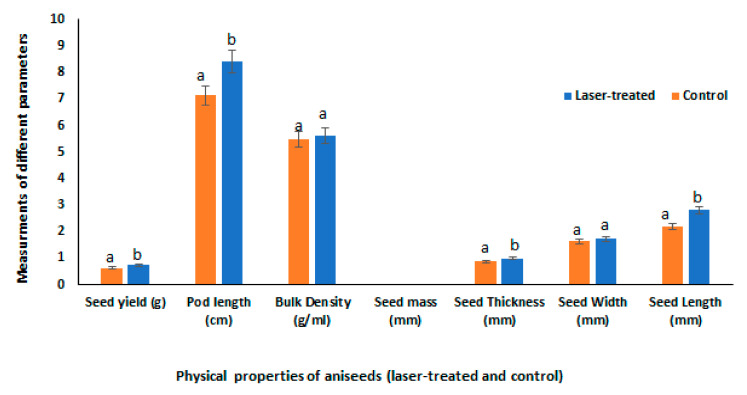
Physical properties in control and laser light-treated anise fruits. Data are represented by the means of at least 3 replicates ± standard error. Different small letters (a, b) above columns indicate significant differences between control and laser-treated samples at *p <* 0.05.

**Figure 2 plants-10-02591-f002:**
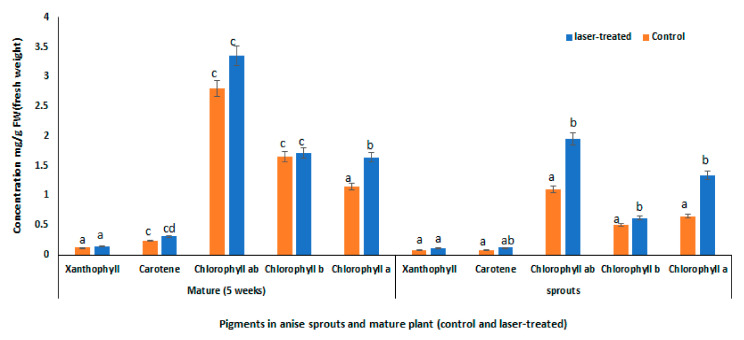
Pigments in sprouts and mature anise (control and laser light-treated). Data are represented by the means of at least 3 replicates ± standard error. Different small letters (a, b, c and d) above columns indicate significant differences between control and laser-treated samples at *p <* 0.05.

**Figure 3 plants-10-02591-f003:**
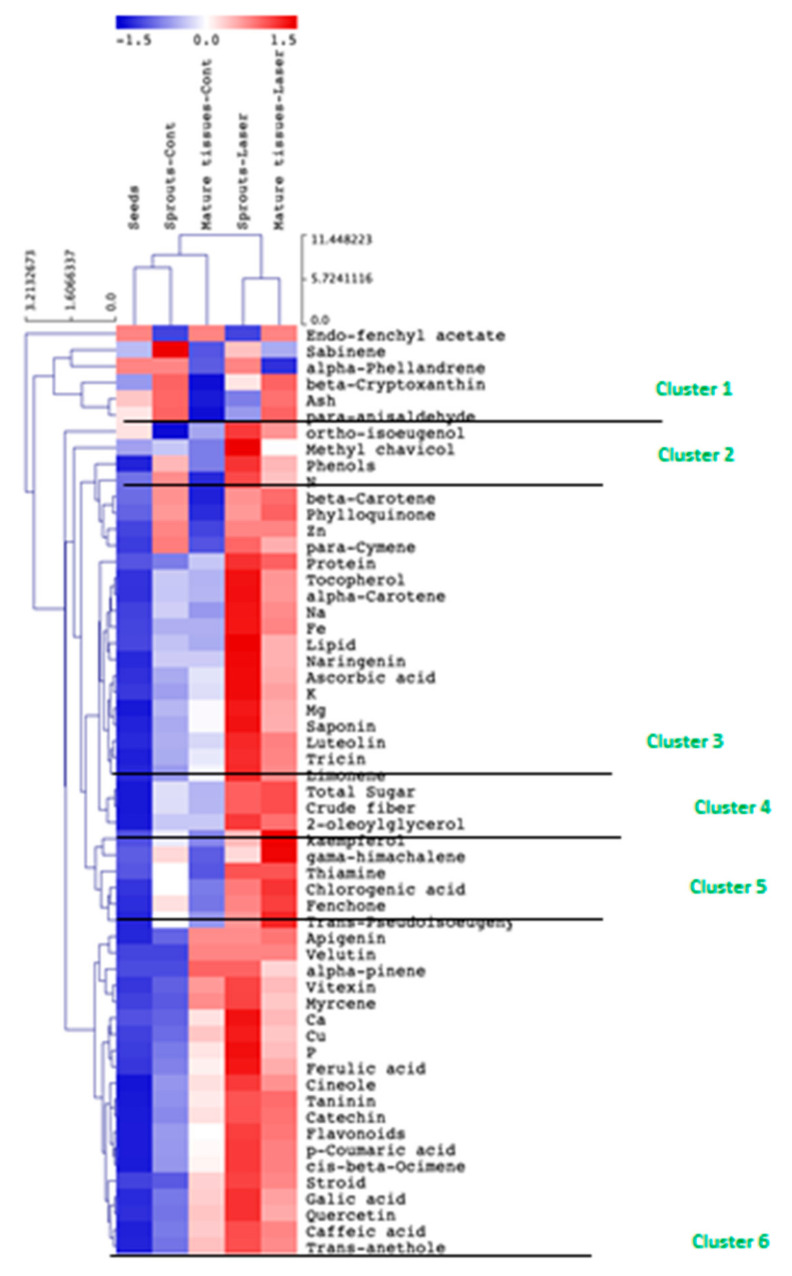
Stage-specific responses of anise fruits, sprouts, and mature plants to the effect of laser light treatment on the nutritional and health-promoting properties. The measured parameters are represented by contents of pigments, total nutrients, minerals, vitamins, essential oils, and phenolic compounds. Data are represented by the means of at least 3 replicates.

**Table 1 plants-10-02591-t001:** Total metabolite contents, minerals, vitamins, and essential oil levels in fruits, sprouts, and mature anise (control and laser light-treated). Data are represented by the means of at least 3 replicates ± standard error. Different small letters (a, b, c, d) within a row indicate significant differences between control and laser-treated samples at *p <* 0.05.

Metabolite	Fruits	Sprouts Control Laser-Treated		Mature Control Laser-Treated	
**Total primary metabolite (mg/g FW)**					
Lipid	84.7 ± 13.7a	109.1 ± 9.7a	173.2 ± 5.9b	104.4 ± 18.3b	138.2 ± 9.9c
Protein	163 ± 19.9a	171.4 ± 10.7a	239.5 ± 4.4b	185.9 ± 26a	230.3 ± 14.4b
Total Sugar	192.4 ± 26.8a	247.9 ± 8.5a	304 ± 31.6b	235.9 ± 34.1b	309.3 ± 19.3c
Ash	2.4 ± 0.16a	2.6 ± 0.2a	2 ± 0.4a	1.8 ± 0.93a	2.58 ± 0.16a
Crude fiber	2.02 ± 0.28a	2.6 ± 0.1a	3.2 ± 0.5b	2.48 ± 0.36ab	3.25 ± 0.2b
**Total secondary metabolite (mg/g FW)**					
Phenols	6.67 ± 0.9a	10.1 ± 0.9a	11.6 ± 0.1b	7.66 ± 1.2a	10.1 ± 0.76a
Flavonoids	0.26 ± 0.04a	0.33 ± 0a	0.5 ± 0b	0.39 ± 0.05a	0.47 ± 0.03a
Taninin	41 ± 6a	47.33 ± 0.4a	63.3 ± 6.2b	55 ± 7a	62 ± 4.5a
Saponin	4.62 ± 0.82a	7 ± 0.4a	12.9 ± 1.2b	8.51 ± 1.23b	10.15 ± 0.7c
Stroid	8.8 ± 1a	9.1 ± 0.6a	13.9 ± 0.7b	12 ± 2b	13 ± 1bc
**Vitamins (mg/g FW)**					
Tocopherol (Vit E)	0.49 ± 0.07a	0.63 ± 0.1a	0.91 ± 0.1b	0.61 ± 0.09a	0.79 ± 0.05ab
α-Carotene (Vit A)	0.28 ± 0.04a	0.36 ± 0.1a	0.52 ± 0.1b	0.35 ± 0.05a	0.45 ± 0.03a
β-Carotene (Vit A)	0.09 ± 0.01a	0.16 ± 0a	0.16 ± 0a	0.07 ± 0.02a	0.17 ± 0.01b
β-Cryptoxanthin (Vit A)	0.05 ± 0.01a	0.07 ± 0a	0.06 ± 0a	0.04 ± 0.01a	0.07 ± 0b
Thiamine (Vit B)	0.05 ± 0.01a	0.06 ± 0a	0.07 ± 0a	0.05 ± 0.01a	0.07 ± 0a
Phylloquinone (Vit K)	0.09 ± 0.01a	0.14 ± 0a	0.14 ± 0a	0.08 ± 0.01a	0.15 ± 0.01b
Ascorbic Acid (Vit C)	1.2 ± 0.2a	1.4 ± 0.1a	1.96 ± 0.1ab	1.5 ± 0.2a	1.7 ± 0.1ab
**Minerals (mg/g DW)**					
K	10.2 ± 1.4a	11.7 ± 1a	16.89 ± 1b	12.7 ± 1.7a	14.7 ± 0.9b
P	3 ± 0.5a	3.46 ± 0.4a	6.28 ± 0.5b	4.7 ± 0.6ab	5 ± 0.3ab
Ca	2.2 ± 0.3a	2.24 ± 0.3a	3.19 ± 0.5b	2.7 ± 0.3a	2.8 ± 0.2a
Mg	0.8 ± 0.17a	1.39 ± 0.5a	2.55 ± 0.4ab	1.65 ± 0.23b	2.01 ± 0.13c
Na	0.32 ± 0.04a	0.37 ± 0.1a	0.47 ± 0.1b	0.35 ± 0.05a	0.43 ± 0.03a
Fe	0.14 ± 0.02a	0.15 ± 0.02a	0.18 ± 0.02a	0.15 ± 0.02a	0.17 ± 0.01a
Zn	0.02 ± 0a	0.04 ± 0.01a	0.04 ± 0.01a	0.02 ± 0a	0.04 ± 0b
Cu	0.07 ± 0.01a	0.08 ± 0.02a	0.17 ± 0.01b	0.13 ± 0.02a	0.13 ± 0.01a
N	24.5 ± 2.9a	32.51 ± 3a	34.58 ± 3a	22.4 ± 3.6a	30.9 ± 2.2b
**Essential oils (mg/g FW)**					
α-pinene	0.02 ± 0a	0.02 ± 0.01a	0.05 ± 0.02b	0.05 ± 0.01b	0.04 ± 0b
Sabinene	0.26 ± 0.03a	0.46 ± 0.22a	0.34 ± 0.07a	0.2 ± 0.04a	0.25 ± 0.03a
Myrcene	0.13 ± 0.03a	0.15 ± 0.03a	0.45 ± 0.09b	0.39 ± 0.05b	0.34 ± 0.02b
Fenchone	3.3 ± 0.43a	4.38 ± 0.67a	4.79 ± 0.75a	3.61 ± 0.55a	5.09 ± 0.32b
p-cymene	0.47 ± 0.06a	0.61 ± 0.18a	0.62 ± 0.1a	0.48 ± 0.07a	0.59 ± 0.04ab
o-isoeugenol	3.9 ± 0.36ab	2.97 ± 0.6a	4.47 ± 0.7b	3.47 ± 0.48a	4.17 ± 0.26ab
1,8-cineole	1.24 ± 0.18a	1.41 ± 0.2a	1.84 ± 0.4a	1.61 ± 0.21ab	1.72 ± 0.11ab
Cis-β-ocimene	0.26 ± 0.04a	0.33 ± 0.04a	0.51 ± 0.11b	0.4 ± 0.05a	0.47 ± 0.03a
Aα-phellandrene	0 ± 0a	0 ± 0a	0 ± 0a	00.14 ± 0a	0.012 ± 0a
Methyl chavicol	0.18 ± 0.03a	0.2 ± 0.04a	0.39 ± 0.08 b	0.16 ± 0.04a	0.23 ± 0.02ab
Endo-fenchyl acetate	0 ± 0a	0.001 ± 0a	0.002 ± 0b	0 ± 0a	0 ± 0a
p-anisaldehyde	0.07 ± 0.01ab	0.08 ± 0.04a	0.06 ± 0.01a	0.05 ± 0.01a	0.08 ± 0a
Limonene	0.12 ± 0.02a	0.18 ± 0.03a	0.35 ± 0.05b	0.23 ± 0.03ab	0.3 ± 0.02c
Stearic acid	0.03 ± 0a	0.03 ± 0.01a	0.03 ± 0.01a	0.03 ± 0a	0.03 ± 0a
2-oleoylglycerol	0.19 ± 0.03a	0.29 ± 0a	0.44 ± 0.05b	0.29 ± 0.04a	0.41 ± 0.03a
γ-himachalene	0.1 ± 0.01a	0.11 ± 0.03a	0.11 ± 0.01a	0.1 ± 0.01a	0.12 ± 0.01a
Trans-Pseudoisoeugenyl	0.13 ± 0.02a	0.15 ± 0.06a	0.16 ± 0.01a	0.14 ± 0.02a	0.17 ± 0.01b
Trans-anethole	44 ± 7a	49.99 ± 8.36a	75.17 ± 2.4b	66.4 ± 8b	70.5 ± 4bc
**Amino acids**					
Phenylalanine	2.0 ± 0.1a	3.2 ± 0.12b	5.8 ± 0.8d	2.8 ± 0.07b	4.2 ± 0.4c
L-phenylalanine aminolyase	25.9 ± 1.8a	33.1 ± 2.1a	57.1 ± 5.4c	28.3 ± 3.1a	49.2 ± 5.41b
DAHPS	0.1 ± 0.02a	0.44 ± 0b	0.95 ± 0.01c	0.41 ± 0b	0.8 ± 0.05c
**Other related compounds**					
Cinnamic acid	2.8 ± 0.1b	1.6 ± 0a	2.7 ± 0.1b	3.1 ± 0.1b	4.07 ± 0.2c
Shikimic acid	33.9 ± 1.1a	58.3 ± 6.8b	74 ± 4.0c	40.5 ± 6.2a	60.8 ± 7.4b

**Table 2 plants-10-02591-t002:** Phenolic and flavonoid profile of fruits, sprouts, and mature anise (control and laser light-treated). Data are represented by the means of at least 3 replicates ± standard error. Different small letters (a, b, c) within a row indicate significant differences between control and laser-treated samples at *p <* 0.05.

		Sprouts		Mature	
Compound	Fruits	Control	Laser-Treated	Control	Laser-Treated
**Phenolic acids (μg/g DW)**					
Caffeic acid	3.86 ± 0.59a	4.41 ± 0.2a	6.44 ± 0.1b	5.64 ± 0.73ab	6.08 ± 0.38b
Ferulic acid	0.03 ± 0.01a	0.04 ± 0a	0.09 ± 0c	0.06 ± 0.01b	0.07 ± 0b
Catechin	1.17 ± 0.17a	1.34 ± 0.3a	1.82 ± 0a	1.59 ± 0.21a	1.77 ± 0.11a
Galic acid	3.76 ± 0.71a	4.84 ± 0.3a	9.72 ± 0.2b	7.54 ± 0.98ab	8.16 ± 0.51b
p-Coumaric acid	1.02 ± 0.16a	1.31 ± 0.1a	2.04 ± 0a	1.58 ± 0.22a	1.88 ± 0.8a
**Flavonoids (μg/g DW)**					
kaempferol	0.44 ± 0.06a	0.82 ± 0a	1.05 ± 0.2a	0.57 ± 0.1a	1.5 ± 0.07a
Chlorogenic acid	0.11 ± 0.01a	0.14 ± 0a	0.16 ± 0a	0.12 ± 0.02a	0.17 ± 0.01a
Quercetin	1.54 ± 0.3a	1.98 ± 0.1a	4.28 ± 0.1b	3.32 ± 0.4ab	3.51 ± 0.2ab
Luteolin	0.04 ± 0.01a	0.07 ± 0a	0.14 ± 0b	0.08 ± 0.01a	0.12 ± 0.01ab
Apigenin	0.16 ± 0.03a	0.21 ± 0a	0.44 ± 0b	0.44 ± 0.04b	0.46 ± 0.02b
Naringenin	0.78 ± 0.14a	1.44 ± 0.1b	2.67 ± 0.1c	1.44 ± 0.23b	2.0 ± 0.14ab
Velutin	0.01 ± 0a	0.01 ± 0a	0.02 ± 0b	0.02 ± 0a	0.02 ± 0a
Tricin	0.77 ± 0.13a	1.17 ± 0.1a	2.06 ± 0ab	1.36 ± 0.2a	1.81 ± 0.11a
vitexin	0.51 ± 0.1a	0.58 ± 0a	1.21 ± 0b	1.06 ± 0.13a	1 ± 0.06a

**Table 3 plants-10-02591-t003:** Antioxidant, anti-lipidemic, and anti-hemolytic activities in fruits, sprouts, and mature anise (control and laser-treated). Data are represented by the means of at least 3 replicates ± standard error. Different small letters (a, b, c) within a row indicate significant differences between control and laser-treated samples at *p* < 0.05.

Activity	Fruits	Sprouts Control Laser-Treated		Mature Control Laser-Treated	
**Antioxidant**					
DPPH (%)	36.5 ± 0.37a	50.2 ± 2.956a	65.61 ± 2.6b	37.41 ± 0.4a	54.27 ± 0.2b
FRAP	6.4 ± 2.3a	15.3 ± 0.6a	27.6 ± 1.9b	14.18 ± 3.1b	24.05 ± 1.5c
Total antioxidant capacity (TAC) (nmol/g FW)	9.39 ± 1.3a	10.72 ± 0.061a	13.23 ± 0.2b	11.5 ± 1.5a	13.4 ± 0.84b
Anti-lipid peroxidation	2.5 ± 0.37a	3.22 ± 0.275a	4.39 ± 0.07b	3.41 ± 0.4b	4.27 ± 0.2c
(ORAC)	571 ± 80a	865.7 ± 34.3a	1064.6 ± 85.5b	821 ± 65b	1082 ± 68c
% inhibation of LDL oxidation (TBARS)	13 ± 2a	14.3 ± 0.9a	28.6 ± 3.7b	24 ± 2ab	25 ± 3b
% inhibation of LDL oxidation (conjugated dienes)	15 ± 3a	15.4 ± 0.3a	28.5 ± 3.4b	17 ± 3a	25 ± 2b
**Anti-lipidemic**					
Anti-Amylase IC_50_(mg/mL)	3.1 ± 0.2a	2.7 ± 0.1a	1.5 ± 0.1b	2.96 ± 0.22a	2.35 ± 0.1b
Anti-Lipase IC_50_(mg/mL)	1.32 ± 0.1a	1.7 ± 0.3a	0.97 ± 0b	1.75 ± 0.14a	1.0 ± 0.1b
Anti-chlostrol	39 ± 5.77a	50.2 ± 1.9a	68.7 ± 3.7b	43 ± 7.5a	67 ± 4.17b
**Anti-hemolytic activity**					
% inhibation of hemolysis	11 ± 1.78a	14 ± 1.1a	22.6 ± 1.8b	18 ± 2.38a	21 ± 1.28a

## Data Availability

Data presented in this study are available on reasonable request.

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
