# Peer review of "Developmental Stages-Specific Response of Anise Plants to Laser-Induced Growth, Nutrients Accumulation, and Essential Oil Metabolism"

_plants, 2021, doi:10.3390/plants10122591_

Round 1

Reviewer 1 Report

Seeds from Pimpinella anisum (anise) are used in human food preparation. Their laser irradiation treatment was the source of oxidation. Together biomass and the level of bioactive compunds was enhanced by laser-induced damage (oxidation).

Lines 17-33

Abstract : The sentence »Mean-while, seed maturation has led to decreased levels of essential oils and their related precursors (e.g., phenylalanine), which were enhanced under laser treatment. » is quite confusing. For example :

« The normal decline in essential oils and related precursors (e.g., phenylalanine) during seed maturation was interrupted by laser exposure.  Likewise, laser treatment also promoted O–methyltransferase and 3-Deoxy-D-arabino-heptulosonate-7-phosphate syn-

thase (DAHPS) activity.»  

I do not understand the last two sentences of the abstract. Are you saying : The boosting of these metablites by laser treatement was developmentally restricted to P. anisum shoots, not stems or seeds.  

Introduction

Line 50-53

Previous reports have demonstrated the positive impact of laser light on increasing the quality and productivity of many plants, ...etc. Productivity is a very crop specific notion. Do you me net accumulation of bioactive products ? I think of mass yield when I think about productivity of crops.

About line 64 : You are discussing various organic compounds in a positive light, however are any negative to humans (carcinogens, antidiuretics, etc) ? I do not know, but perhaps trace doses can be benefical and avert potential negative affects ?

Line 69 : « To the best of our » is a new paragraph.

Line 86 : It might be helpful to explain to the reader why you chose those laser parameters ?

Line 111 : In this analysis are you using the previously mentioned acetone extracts ? Are you using acid hydrolyzed dry matter ? How do you go from plant samples into the ICP/MS ?

Results and Discussion

Line 299 : « Generally, the yield and contents of essential oil of plant material may

vary depending on many factors such as temperature, radiation, soil fertility, plant part

used and harvesting time [13]. » This seems out of place after talking about essential oils and headaches.  

Line 395-end of Discussion : This paragraph on really tissue specific results either needs to be deleted or rewritten to allow the reader to appreciate this is not an unusual result. Plant physiology (this sink/source) often partitions metabolites and/or potential toxicants in less vegetative tissues, often excluding them from tissues associated with reproduction. Because « sprouts » are undergoing a great deal of vegetative development, their inducibility may reflect windows of adaptation to stress unavailable in the other non-inducing tissues ?

Please delete « as an ecofriendly approach » from your conclusions. While this word is a popular press word, I believe you mean « advantageous approach ». I am seeing speciality facilities that link laser exposure to primary and secondary metabolite production.  

Overall this is a very interesting paper, and the authors represent a great collaboration to reveal a fascinating overlap between the biological and physical sciences. I would like to see corrections or comments about the above comments, and subsequently I am hoping to strongly endorse the publication of this work after the corrections are made.

Author Response

­Dear Editor,

Thank you for your comments on our article manuscript entitled "Developmental stages-specific response of anise plants to laser-induced growth, nutrients accumulation, and essential oil metabolism". We have carefully read all comments and suggestions and made as much modifications as we believed to meet the reviewers’ requests. A detailed point by point response to reviewers’ comments is addressed at the end of this letter (typed in red). 

Reviewer 1

Seeds from Pimpinella anisum (anise) are used in human food preparation. Their laser irradiation treatment was the source of oxidation. Together biomass and the level of bioactive compunds was enhanced by laser-induced damage (oxidation).

Lines 17-33

Abstract : The sentence »Mean-while, seed maturation has led to decreased levels of essential oils and their related precursors (e.g., phenylalanine), which were enhanced under laser treatment. » is quite confusing. For example :

« The normal decline in essential oils and related precursors (e.g., phenylalanine) during seed maturation was interrupted by laser exposure.  Likewise, laser treatment also promoted O–methyltransferase and 3-Deoxy-D-arabino-heptulosonate-7-phosphate synthase (DAHPS) activity.»  

 I do not understand the last two sentences of the abstract. Are you saying : The boosting of these metablites by laser treatement was developmentally restricted to P. anisum shoots, not stems or seeds.  

Response: Thanks for valuable comments; we have edited these sentences for more clarification

Introduction

Line 50-53

Previous reports have demonstrated the positive impact of laser light on increasing the quality and productivity of many plants, ...etc. Productivity is a very crop specific notion. Do you me net accumulation of bioactive products? I think of mass yield when I think about productivity of crops.

Response: Thanks, actually we mean the accumulation of bioactive metabolites, so we corrected this in the manuscript

About line 64: You are discussing various organic compounds in a positive light, however are any negative to humans (carcinogens, antidiuretics, etc) ? I do not know, but perhaps trace doses can be benefical and avert potential negative affects ?

Response: According to many reports, these kinds of light should be used in a certain dose, otherwise it might have harmful effects on humans.   

Line 69 : « To the best of our » is a new paragraph.

Response: Thanks, done 

Line 86 : It might be helpful to explain to the reader why you chose those laser parameters ?

Response: According to a preliminary experiment and our previous studies (as we mentioned in line 88, the selected parameters were tested for their ability to induce germination and plant growth, which would be reflected on increasing the plant bioactive content.

Line 111 : In this analysis are you using the previously mentioned acetone extracts ? Are you using acid hydrolyzed dry matter ? How do you go from plant samples into the ICP/MS ?

Response: Thanks, more details were added to this section for clarification  

Results and Discussion

 Line 299 : « Generally, the yield and contents of essential oil of plant material may vary depending on many factors such as temperature, radiation, soil fertility, plant part used and harvesting time [13]. » This seems out of place after talking about essential oils and headaches. 

Response: Thanks, we deleted this sentence.  

Line 395-end of Discussion: This paragraph on really tissue specific results either needs to be deleted or rewritten to allow the reader to appreciate this is not an unusual result. Plant physiology (this sink/source) often partitions metabolites and/or potential toxicants in less vegetative tissues, often excluding them from tissues associated with reproduction. Because « sprouts » are undergoing a great deal of vegetative development, their inducibility may reflect windows of adaptation to stress unavailable in the other non-inducing tissues?

Response: Thanks, we adjusted and extended this section

Please delete « as an ecofriendly approach » from your conclusions. While this word is a popular press word, I believe you mean « advantageous approach ». I am seeing speciality facilities that link laser exposure to primary and secondary metabolite production.   

Response: Thanks, we replaced (ecofriendly) by (advantageous). 

Overall this is a very interesting paper, and the authors represent a great collaboration to reveal a fascinating overlap between the biological and physical sciences. I would like to see corrections or comments about the above comments, and subsequently I am hoping to strongly endorse the publication of this work after the corrections are made.

Reviewer 2 Report

I find this paper very interesting and well written. I have only few concerns about the content.

1. What is the unit in Table 1 for “Essential oils”?

2. In “essential oils” there is PAL and DAHPS, what are the units in this case? Are those activities, protein amounts, how was it determined? Should not the enzymes be separated from metabolites in the Table?

3. Minor typos and editorial mistakes like in line 89: “…seeds  were  and  rinsed  in  distilled  water…” or line 307: “…being converted into chorismiate…”.

Author Response

­Dear Editor,

Thank you for your comments on our article manuscript entitled "Developmental stages-specific response of anise plants to laser-induced growth, nutrients accumulation, and essential oil metabolism". We have carefully read all comments and suggestions and made as much modifications as we believed to meet the reviewers’ requests. A detailed point by point response to reviewers’ comments is addressed at the end of this letter (typed in red).

Reviewer 2

I find this paper very interesting and well written. I have only few concerns about the content.

  1. What is the unit in Table 1 for “Essential oils”?

Response: Thanks, we inserted it in the table

  1. In “essential oils” there is PAL and DAHPS, what are the units in this case? Are those activities, protein amounts, how was it determined? Should not the enzymes be separated from metabolites in the Table?

Response: Thanks, we added more details about determination of PAL and DAHPS activities. These enzymes are mainly involved in biosynthesis of several metabolites; therefore, we included them in table with essential oils in one table.

  1. Minor typos and editorial mistakes like in line 89: “…seeds  were  and  rinsed  in  distilled  water…” or line 307: “…being converted into chorismiate…”.

Response: Thanks, we checked the whole manuscript to correct all mistakes.

Reviewer 3 Report

Review summary

Article is curious, however, some things should be explained and corrected before publication. They are listed below:

Major issues:

- Authors are sure, that anise seeds were investigated? Generally anise is usually used as whole fruits than separated seeds.

- Propose add additional section “material and reagents” and describe all used chemicals and reagents. In some paragraph they were describe, but not always.

- Essential oils analysis should be more accurate described. Authors should describe GC-MS equipment and GC-MS column.

- Results of hierarchical fuzzy clustering should be more accurate described. Moreover Figure 3 is unclear. Please correct it.

- Results of GC-MS analysis with retention index and HPLC analysis with retention time should be presented.

Minor issues:

Line 61-62 – Please remember that Anisi fructus is mainly carminativum and antispasmoliticum remedies.

Author Response

­Dear Editor,

Thank you for your comments on our article manuscript entitled "Developmental stages-specific response of anise plants to laser-induced growth, nutrients accumulation, and essential oil metabolism". We have carefully read all comments and suggestions and made as much modifications as we believed to meet the reviewers’ requests. A detailed point by point response to reviewers’ comments is addressed at the end of this letter (typed in red).

Reviewer 3

Article is curious, however, some things should be explained and corrected before publication. They are listed below:

 Major issues:

- Authors are sure, that anise seeds were investigated? Generally anise is usually used as whole fruits than separated seeds.

Response: Thanks, we have made sure that the fruit was investigated, not the seed, so we corrected this in the whole manuscript

- Propose add additional section “material and reagents” and describe all used chemicals and reagents. In some paragraph they were describe, but not always.

Response: Thanks, We described the sources of chemicals and reagents used in our research.

- Essential oils analysis should be more accurate described. Authors should describe GC-MS equipment and GC-MS column.

Response: Thanks, we added more details about GC-MS analysis of essential oils.

- Results of hierarchical fuzzy clustering should be more accurate described. Moreover Figure 3 is unclear. Please correct it.

Response: Thanks, we added more details about the hierarchical clustering analysis and also we corrected the figure.

Round 2

Reviewer 3 Report

Article was improved, however some things have to be corrected.

Major issues:

- I am not a native speaker, but I think, that language of the article needs some attention.

- Please divide section 2.6 into several subsections (GC-MS analysis, Determination of PAL, Determination of DAHPS).

- Please divide section 2.7 into two sections (HPLC analysis, Determination of Total Phenolic content)

Supplementary table 1 – Please divide table into two tables (first for GC-MS and second for HPLC) or add RT of polyphenols to the table 2.

GC-MS table should also contain relative peaks area. I think, that authors should also add GC-MS and HPLC chromatogram figures to the supplementary materials.

- Section 3.5. If possible authors should discus own results with literature. Did obtain results expected or unexpected?

- 2.9. Statistical analyses. Authors should describe how they performed hierarchical clustering analysis. There no information about this anywhere in material and methods section.

Minor issues:

Line 202-203 – “For retention index (RI) of identified 203 essential oil, please check Supplementary Table 1.” This is about essential oils, but not polyphenols. Please correct the line and describe how polyphenols was identified? Comparison with standard? Comparison with known UV spectra and literature?

Author Response

­Dear Editor,

Thank you for your comments on our article manuscript entitled "Developmental stages-specific response of anise plants to laser-induced growth, nutrients accumulation, and essential oil metabolism". We have carefully read all comments and suggestions and made as much modifications as we believed to meet the reviewers’ requests. A detailed point by point response to reviewers’ comments is addressed at the end of this letter (typed in red). 

Reviewer 1

 Major issues:

- I am not a native speaker, but I think, that language of the article needs some attention.

Response: Thanks; we carefully checked the English language in the whole manuscript.

- Please divide section 2.6 into several subsections (GC-MS analysis, Determination of PAL, Determination of DAHPS).

Response: Thanks; done

- Please divide section 2.7 into two sections (HPLC analysis, Determination of Total Phenolic content)

Response: Thanks; done

Supplementary table 1 – Please divide table into two tables (first for GC-MS and second for HPLC) or add RT of polyphenols to the table 2.

Response: Thanks; we divided supplementary table 1 into two tables (first for GC-MS and second for HPLC)

GC-MS table should also contain relative peaks area. I think, that authors should also add GC-MS and HPLC chromatogram figures to the supplementary materials.

Response: Thanks; we added relative peaks area in the supplementary tables, but we thought that it is not required to put them in the table.

We already added these chromatogram figures to the supplementary materials.

- Section 3.5. If possible authors should discus own results with literature. Did obtain results expected or unexpected?

Response: Thanks for valuable comment; we have discus our own results with literature in this section. The obtained results were expected according to some previous studies. 

- 2.9. Statistical analyses. Authors should describe how they performed hierarchical clustering analysis. There no information about this anywhere in material and methods section.

Response: Thanks, we added hierarchical clustering analysis to this this section

Minor issues:

Line 202-203 – “For retention index (RI) of identified 203 essential oil, please check Supplementary Table 1.” This is about essential oils, but not polyphenols. Please correct the line and describe how polyphenols was identified? Comparison with standard? Comparison with known UV spectra and literature?

 Response: Thanks for valuable comment; we actually identified polyphenols by comparison with standards. So, the detection of concentration of each compound was done using a calibration curve of the corresponding standard. For more clarification, we provided examples of calibration curves of some phenolic standards and chromatogram of the target phenolic standards in the supplementary materials.

Round 3

Reviewer 3 Report

Article was great improved. However, there are somethings, that will be corrected before publication.

Major issues:

Please add citation to point 2.2. (photosynthesis rate).

Minor issues:

- Please remove all unwanted spaces and lines.

Line 159 - … AbdElgawad et al., (2019)… Please correct citation style

Author Response

Dear editor,

We would thank you for the thorough review of our manuscript and for the valuable suggestions and comments, which surely will improve the final version of this manuscript.

We have addressed comments and modified the manuscript accordingly. All changes are tracked or yellow-highlighted in the revised manuscript. Below is our point-by-point response (in red bold) to each comment (black).

Major issues:

Please add citation to point 2.2. (photosynthesis rate).

Done as recommended.

Minor issues: -

Please remove all unwanted spaces and lines.

Done as recommended

Line 159 - … AbdElgawad et al., (2019). Please correct citation style.

Done as recommended